# Repetitive Trans Spinal Magnetic Stimulation Improves Functional Recovery and Tissue Repair in Contusive and Penetrating Spinal Cord Injury Models in Rats

**DOI:** 10.3390/biomedicines9121827

**Published:** 2021-12-03

**Authors:** Amandine Robac, Pauline Neveu, Alizée Hugede, Elisabeth Garrido, Lionel Nicol, Quentin Delarue, Nicolas Guérout

**Affiliations:** 1EA3830 GRHV, Institute for Research and Innovation in Biomedicine (IRIB), Normandy University, UNIROUEN, 76000 Rouen, France; amandine.robac1@univ-rouen.fr (A.R.); pauline.neveu2@gmail.com (P.N.); hugede.alizee@gmail.com (A.H.); elisabethgarrido66@gmail.com (E.G.); quentin.delarue@univ-rouen.fr (Q.D.); 2Department of Neurosurgery, Rouen University Hospital, 1 Rue de Germont, 76000 Rouen, France; 3Institute for Research and Innovation in Biomedicine (IRIB), Normandy University, UNIROUEN, Inserm U1096, 76000 Rouen, France; lionel.nicol@univ-rouen.fr

**Keywords:** rehabilitation, spinal cord injury, glial scar, magnetic stimulation, cystic cavities and functional recovery

## Abstract

Spinal cord injury (SCI) is an incurable condition in which the brain is disconnected partially or completely from the periphery. Mainly, SCIs are traumatic and are due to traffic, domestic or sport accidents. To date, SCIs are incurable and, most of the time, leave the patients with a permanent loss of sensitive and motor functions. Therefore, for several decades, researchers have tried to develop treatments to cure SCI. Among them, recently, our lab has demonstrated that, in mice, repetitive trans-spinal magnetic stimulation (rTSMS) can, after SCI, modulate the lesion scar and can induce functional locomotor recovery non-invasively. These results are promising; however, before we can translate them to humans, it is important to reproduce them in a more clinically relevant model. Indeed, SCIs do not lead to the same cellular events in mice and humans. In particular, SCIs in humans induce the formation of cystic cavities. That is why we propose here to validate the effects of rTSMS in a rat animal model in which SCI leads to the formation of cystic cavities after penetrating and contusive SCI. To do so, several techniques, including immunohistochemical, behavioral and MRI, were performed. Our results demonstrate that rTSMS, in both SCI models, modulates the lesion scar by decreasing the formation of cystic cavities and by improving axonal survival. Moreover, rTSMS, in both models, enhances functional locomotor recovery. Altogether, our study describes that rTSMS exerts positive effects after SCI in rats. This study is a further step towards the use of this treatment in humans.

## 1. Introduction

Spinal cord injury (SCI) is a debilitating condition which can lead to a permanent loss of motor and sensitive functions. To date, there is no curative treatment which can be proposed to injured patients. That is why fundamental and preclinical studies have been conducted over the last several decades to find innovative therapies [1]. Research studies have followed different paths; the first one tried to modulate the inhibitory microenvironment present after SCI, and the second one harnessed to replace the lost cells, such as neurons or oligodendrocytes [2]. These two strategies are based on the knowledge acquired about the cellular and molecular events that take place after SCI. Indeed, at the cellular level, SCI induces a massive inflammation with the infiltration of circulating immune cells, such as macrophages, and induction of microglia reactivity [3]. At the same time, the initial traumatic injury leads to neuronal and oligodendroglial death [2,4]. Altogether, these cellular events lead to the formation of a spinal scar composed of a fibrotic core that is present in the lesion epicenter and of an astroglial scar located at the border of the lesion [5,6,7]. This spinal scar exerts complementary and opposite effects. In fact, it segregates the inflammatory cells to avoid the expansion of the lesion, but it also inhibits axonal regrowth. In effect, the spinal scar secretes a wide range of extracellular matrix molecules, including chondroitin sulfate proteoglycan, a class of molecules known as one of the major inhibitory components of the lesion scar [8]. To alleviate this inhibitory microenvironment, we and other research groups have assessed the effects of different therapies, such as cellular transplantation, biomaterials’ graft or more recently repetitive magnetic stimulation [9,10,11,12,13]. To do so, rodent models have been massively used, mainly mice and rats. However, it is important to note that the cellular events and also the composition of the spinal scar that take place after SCIs are distinct between mice and rats but also with humans. In effect, in rats and humans, the fibrotic core present at the lesion epicenter of the spinal scar in mice is replaced progressively by cystic cavities within weeks to months after SCI [14]. That is why it can be of primary importance, before we translate an innovative therapy to clinical practice in humans, to validate it in mice and also in rats. In a clinically related perspective, it is also important to mimic the lesion which occurs in humans. Indeed, in humans, SCIs are mainly due to contusive injury, whereas, in preclinical models, SCIs are often penetrating, because this model is reproducible and does not require specific apparatuses [15].

Recently, our research group reported that repetitive trans-spinal magnetic stimulation (rTSMS) can be used as a non-invasive treatment after SCI. Our study demonstrates that rTSMS modulates the spinal scar and enhances axonal regrowth and locomotor recoveries [9,16,17]. These experiments were conducted in a mouse model, and SCI was performed by complete transection. That is why we propose to validate this promising therapy in a rat model. Our work was conducted in both penetrating and contusive SCI in a large cohort of 104 rats.

## 2. Materials and Methods

### 2.1. Animal Care and Use Statement

The protocols were designed to minimize pain or discomfort to the animals. All experimental procedures were in accordance with the European Community guiding principles on the care and use of animals (86/609/CEE; Official Journal of the European Communities no. L358; 18 December 1986), French Decree no. 97/748 of 19 October 1987 (Journal Officiel de la République Française; 20 October 1987), and the recommendations of the Cenomexa ethics committee (#23767).

Experiments were performed on adult Sprague Dawley female rats (Janvier Labs, Le Genest-Saint-Isle, France) at eight–ten weeks of age (average weight 260–280 g).

Female rats were housed (two rats/cage) in secure conventional rodent facilities on a 12-h light/dark cycle, with constant access to food and water.

Our two procedures were composed of two main experimental groups:

SCI control group: animals received SCI.

SCI + STM group: animals received SCI and rTSMS treatment during 15 days.

For the two procedures, histological analyses were performed at 15 and 60 days after SCI. Functional analyses, including plantar and locotronic tests, were performed at 15, 30 and 60 days after SCI.

For the second procedure only (contusive SCI), MRI experiments were performed at 7, 21 and 42 days after SCI.

A total of 104 rats were included in the entire study (Procedure 1 and Procedure 2).

A general overview of these experiments is presented in Figure 1.

### 2.2. Experimental Design

#### 2.2.1. Procedure 1

##### Surgical Procedure

In order to assess the effects of the rTSMS treatment and to compare them to those described in mice, first we performed penetrating SCI by using a 25-gauge needle in rats, as described previously [9,16].

For Procedure 56, rats were used (Figure 1):- Two groups of 8 rats for the immunohistological experiments 15 days after SCI;- Two groups of 20 rats for functional tests and for immunohistological experiments 60 days after SCI.

#### 2.2.2. Procedure 2

##### Surgical Procedure

In humans, SCIs are mainly caused by contusive injuries; thus in a second experiment, we investigated the effects of rTSMS in a contusion model in rats. A moderate/severe lesion model was chosen due to the fact that 60% of the lesions in humans are incomplete [14].

The surgery was performed as described above, except for the lesion step.

Indeed, after laminectomy, the dura mater was removed, and contusion injury was applied by using a force-controlled spinal cord impactor (IH-0400 Impactor, Precision Systems and Instrumentation LLC, Brimstone, VA, USA). SCIs were performed by applying a force set to 175 kdyn. For Procedure 56, rats were used, as described above for Procedure 1 (Figure 1).

### 2.3. rTSMS Treatment

We applied rTSMS treatment as previously described by Chalfouh et al. [9].

### 2.4. Functional Test

#### 2.4.1. Locotronic Test: Foot Misplacement Apparatus

Experiments were performed as described previously by Chalfouh et al. (Intellibio, Nancy, France) [9].

#### 2.4.2. Hargreaves Apparatus

A Hargreaves plantar test was performed as described previously by Zhang et al. [18]. Baseline parameters were obtained with an additional group of non-injured animals.

### 2.5. MRI Imaging

In vivo imaging experiments were performed to follow on the same rats the evolution of the spinal cord structure over time, after SCI. Analysis of the spinal cord structure was achieved by the MRI BioSpec Advanced II (Bruker, Ettlingen, Mannheim, Germany), with a magnetic field of 4.7 Teslas, and monitored with ParaVision software.

The rats were anesthetized by intraperitoneal injections (Thiopental, 1 g/20 mL, Panpharma). To perform the MRI recordings, the rats were placed in supine position. The vertebra T9 was put down on the marker of the antenna in order to have the area of interest in the field of view of the MRI. Two cardiac electrodes were placed on the rats to follow their cardiac constants. A comprehensive analysis, including T2*-weighted gradient echo in axial and sagittal sections, was carried out with the parameters presented in the Table 1.

The sequences were performed on axial sections is a multi-slice gradient that allowed us to provide a T2* map for tissues characterization.

The sequences of images were then analyzed by using the following software programs:ParaVision software (Paravision 5.1, Bruker Biospin MRI, Ettlingen, Germany) to determine the surface quantification of hyposignal and hypersignal, but also identify tissues structure on sagittal images;Osirix software (OsiriX 11, Pixmeo SARL, Geneva, Switzerland), to determine the volume of spinal cord section and realize a 3D representation on axial images.

### 2.6. Tissue Preparation and Sectioning

Tissue preparation and cryosectioning were performed as previously described [9,16].

### 2.7. Immunohistochemistry

Immunohistochemistry was performed as described previously [9,16]. The following primary antibodies were used: rabbit anti-platelet-derived growth factorβ (PDGFRβ, Abcam, Cambridge, UK, ab32570), mouse anti-glial fibrillary acidic protein (GFAP Cy3-conjugated Sigma-Aldrich, C9205 and GFAP unconjugated Sigma-Aldrich, G3893), rabbit anti-ionized calcium-binding adapter molecule 1 (Iba1, Wako, 019-19741, Osaka, Japan) and mouse anti-neurofilament 200 kD (NF200, Millipore, Burlington, MA, USA, MAB5256).

### 2.8. Image Acquisition Analysis

Representative images were acquired as described previously.

### 2.9. Quantification of Immunohistochemically Stained Areas

On sagittal sections, the GFAP-negative (GFAP-), PDGFRβ-positive (PDGFRβ+), NF200-negative (NF200-) and DAPI-negative (DAPI-) areas were measured at the epicenter of the lesion and section rostral and caudal to the injury site; thus, a minimum of 3 sections (60 μm) per animal were measured. For Iba1 staining, an analysis of the area in which Iba1-positive amyboid cells were present was performed. Analysis of Iba1 intensity measurement was performed on rectangle of 6 μm × 2 μm. Iba1 intensities were collected after threshold standardization.

### 2.10. Statistical Analysis

Data are presented as means ± standard deviation (SD). Comparisons of means were performed by using two-tailed Mann–Whitney tests for all the experiments. In all tests, *p* < 0.05 was considered statistically significant.

## 3. Results

### 3.1. Procedure 1

The main aim of the first procedure was to investigate the effects of rTSMS after a complete transection of the spinal cord in rats. We focused this study on functional recovery and tissue repair (Figure 1).

#### 3.1.1. rTSMS Treatment Induces Locomotor Recovery after a Complete Transection of the Spinal Cord in Rats

Based on locotronic and Hargreaves tests, motor and sensitive recoveries were assessed respectively.

Hargreaves test results demonstrate that there is no difference between groups at 15, 30 and 60 days after SCI (Appendix A Appendix A).

Locotronic results demonstrate that, 15 days after SCI, rTSMS treatment did not enhance locomotor abilities (Figure 2A–C); indeed, there is no significant difference between SCI and Stm groups at this time point (Figure 2A–C). In contrast, 30 and 60 days after SCI, rTSMS-treated animals showed a significant improvement of the locomotion (Figure 2D–I). In fact, the Stm group presents a reduction in the number of back-legs errors (Figure 2D,G), the total back-leg errors’ time (Figure 2E,H) and total crossing time (Figure 2F,I) 30 and 60 days after SCI.

#### 3.1.2. rTSMS Treatment Enhances Tissue Repair after a Complete Transection of the Spinal Cord in Rats

In order to investigate the effects of rTSMS on tissue repair, immunohistological experiments were performed 15 and 60 days after SCI (Figure 3 and Figure 4, respectively).

Glial and fibrotic components of the scar were studied (Figure 3A–H). It appears that, 15 days after SCI, rTSMS treatment reduces the fibrotic component of the scar (Figure 3H) but has no major effect at this time point on the glial component of the scar (Figure 3G). In contrast, 60 days after SCI, our results reveal that rTSMS modulates the spinal scar by decreasing the GFAP-negative (GFAP-) area (Figure 4G). At this time point, rTSMS does not exert a major effect on the fibrotic component of the scar (Figure 4H).

In rats, one of the main issue after SCI is the presence of cystic cavities which take place into the epicenter of the lesion. To measure this process, the DAPI-negative (DAPI-) area was assessed at 15 and 60 days after SCI. Fifteen days after SCI, there is no difference between the two groups of animals (Figure 3I,L,O), whereas, at 60 days, rTSMS tends to decrease the size of the cavities (*p* = 0.1135) (Figure 4I,L,O).

After SCI, neuronal death and axonal degeneration impair functional recovery, that is why 15 and 60 days after SCI, axonal quantification was investigated via NF200 staining. Our results show that at both time points; 15 days (Figure 3J,M,P) and 60 days after SCI (Figure 4J,M,P) rTSMS treatment decreases the NF200-negative (NF200-) area.

Finally, inflammatory processes were investigated 15 and 60 days after SCI. Indeed, inflammation and reactivity of the immune cells, mainly microglia and macrophages, are key factors regulating tissue healing. Thus, reactivity of microglia/macrophages was assessed by using Iba1 staining. These measurements reveal that, 15 days (Figure 3R,U,W) and 60 days (Figure 4R,U,W) after SCI, rTSMS treatment decreases the amount of amyboid-Iba1-positive cells into the injured spinal cord parenchyma. Iba1’s intensity was also assessed and did not show any difference between groups 15 and 60 days after SCI (Figure 3X and Figure 4X).

### 3.2. Procedure 2

The main aim of this second procedure was to investigate the effects of rTSMS after a moderate/severe contusion of the spinal cord in rats. In addition to functional recovery and tissue repair, MRI analyses were performed (Figure 1).

#### 3.2.1. rTSMS Treatment Induces Functional Recovery after Contusive SCI in Rats

First, sensitive recovery was investigated. To do so, hindpaw withdrawal thermal latency was measured by using Hargreaves plantar test 15, 30 and 60 days after SCI (Figure 5). These experiments show that, at 15, 30 and 60 days after SCI rTSMS, the treated animals presented a reduction of the hindpaw withdrawal thermal latency in comparison to untreated (SCI) animals (Figure 5A–C).

Then, based on locotronic test, functional recoveries were assessed. Our results demonstrate that, 15 and 30 days after SCI, rTSMS treatment did not enhance locomotor abilities (Figure 6A–F); indeed, there is no significant difference between SCI and Stm groups at these time points (Figure 6A–F). More interestingly, 60 days after SCI, rTSMS-treated animals showed a significant improvement in locomotion (Figure 6G–I). Indeed, the Stm group presented a reduction in the number of back-legs errors (Figure 6G), total back-legs errors’ time (Figure 6H) and total crossing time (Figure 6I) 60 days after SCI.

#### 3.2.2. MRI Analyses Show That rTSMS Treatment Decreases Cystic Cavities and Increases Spinal Cord Spared Tissue

MRI experiments were performed at 7 (Figure 7A–F), 21 (Figure 7G–L) and 42 (Figure 7M–R) days after SCI in order to follow on the same rats the evolution of the injury site overtime after SCI and rTSMS treatment. Hyposignal measurement was representative of the fibrotic/necrotic tissue present into the parenchyma, whereas the hypersignal indicates the presence of inflammation at early time points (7 and 21 days, Figure 7A–L) and cystic cavities at a later time point (42 days, Figure 7M–R). Measurement of both hypo- and hypersignal was also performed, and it reflects the overall area of the injured tissue (Figure 7E,K,Q). Based on this overall area measurement, a ratio of lesioned tissue among the entire spinal cord parenchyma was calculated (Figure 7F,L,R).

The MRI analyses reveal that, 7 and 21 days after SCI, there is no difference between groups (Figure 7A–L). In contrast, 42 days after SCI, rTSMS-treated animals presented a reduction of the hypersignal area (Figure 7M,N,P), the overall area of the injured tissue (Figure 7Q) and the ratio of lesioned tissue (Figure 7R).

#### 3.2.3. rTSMS Treatment Enhances Tissue Repair after Contusive SCI in Rats 60 Days after SCI

In order to investigate the effects of rTSMS on tissue repair, immunohistological experiments were performed 15 and 60 days after SCI (Figure 8 and Figure 9, respectively).

In this model, in the same way as the MRI results, immunohistological analyses demonstrate that, 15 days after SCI, there is no difference between treated (Stm) and untreated (SCI) animals (Figure 8).

At the opposite, 60 days after SCI, the analysis of the glial and fibrotic components of the scar shows that rTSMS treatment decreases the GFAP-negative area without decreasing the PDGFRβ-positive area (Figure 9A–H). In the same way, at this time point, rTSMS treatment decreases cystic cavities (DAPI-negative area) and axonal degeneration (NF200-negative area) (Figure 9I,L,O and J,M and P, respectively). Moreover, rTSMS treated group presents an increase of the amount of amyboid-Iba1-positive cells (Figure 9R,U,W). Iba1 intensity was also assessed and did not show any difference between groups 60 days after SCI (Figure 9X).

## 4. Discussion

The main aim of our study was to investigate the effects of rTSMS in penetrating and contusive SCI models in rats (Figure 1). Indeed, the effects of rTSMS were measured first in a penetrating injury model in which the spinal cord was transected with a needle. This model was chosen because it was used initially in our princeps studies in mice [9,16]. As described in mice, in this model, rTSMS treatment enhances functional recovery and modulates the lesion scar in rats. In effect, 15 days after SCI, treated rats present a reduction of the fibrotic scar and an increase of the axonal survival (reduction of NF200-negative area) (Figure 3). These effects on tissue repair are correlated with functional recovery at later time points: 30 and 60 days after SCI (Figure 2). Finally, 60 days after SCI, treated animals present an improvement of the repair of the spinal cord by the decrease of the size of the cavities and the increase of the glial scar (Figure 4).

In a second time, we assessed the effects of rTSMS in a more clinically relevant model. Indeed, a vast majority of the SCIs in humans are contusive and not penetrating [19]. For this procedure, a moderate/severe contusive SCI model was chosen because, in humans, 60% of the SCIs are not complete [19]. This model induces a severe histological injury characterized by large cystic cavities and moderate locomotor deficits. This procedure completed the first one in which the animals present, in addition to a severe histological injury, a complete paraplegia. To do so, standardized contusive SCI was performed by using a force-controlled spinal cord impactor. In this model, it appears that rTSMS also improves functional recovery and tissue repair. More interestingly, our results indicate that, in this model, rTSMS treatment has a major effect at later time point. In fact, rTSMS-treated animals present a significant functional recovery only 60 days after SCI (Figure 6). In the same way, histological analyses reveal that, 15 days after SCI, there was no difference between the two groups of rats, whereas, 60 days after SCI, treated animals showed a reduction of the size of the cavities and an increase of the glial scar and the neuronal survival (Figure 8 and Figure 9). In addition to these analyses, MRI experiments were also performed 7, 21 and 42 days after SCI (Figure 7). These experiments confirm our histological results. In effect, 7 and 21 days after SCI, there was no difference between the two groups of rats. In contrast, 42 days after SCI, MRI results show that rTSMS-treated animals present a reduction of the hypersignal (corresponding to the cavities), the hypersignal and hyposignal (corresponding to the lesioned tissue) and the ratio of lesioned tissue among the entire spinal cord parenchyma (Figure 7).

Ultimately, in our study, we tried to evaluate the sensitive recovery after SCI. Indeed, the plantar test was performed in both groups of animals and in both procedures, 15, 30 and 60 days after SCI (Figure 1). Based on that test, hindpaw withdrawal thermal latency was recorded. In Procedure 1 (penetrating SCI), plantar test results did not show any difference between groups (Appendix A Appendix A). In contrast, in Procedure 2 (contusive SCI), it appears that rTSMS treated animals present a reduction of hindpaw withdrawal thermal latency recordings for all time points in comparison to untreated animals (Figure 5). More interestingly, a closer analysis of these results allows us to see that, at 15 days after SCI, hindpaw withdrawal thermal latency recordings of the rTSMS-treated animals are shorter in comparison to non-injured animals (dashed line, Figure 5A). At the opposite, at later time points, 30 and 60 days after SCI, hindpaw withdrawal thermal latency recordings of the rTSMS-treated animals are comparable to the ones of non-injured animals (dashed lines, Figure 5B,C). These results can illustrate the fact that, at an early time-point (15 days), rTSMS treatment induces allodynia, due to the shorter latency than non-injured animals, whereas, at later time points (30 and 60 days after SCI), rTSMS enhances axonal survival and re-establishes an efficient voluntary sensitive-motor loops close to those of the uninjured rats. Allodynia is a commonly described side-effect of several therapies after SCI, such as cellular transplantation. In effect, cellular transplantation plays its presumed effects mainly by replacing lost cells or by secreting trophic factors which increase axonal survival or axonal regrowth. These benefits can, in turn, exert detrimental side-effects, such as neuropathic pain or allodynia [20,21]. We can hypothesize that, at an early time point after contusive SCI, rTSMS enhances allodynia at least in the case of thermoalgic stimulations due to enhancement of axonal survival, and that, at later time points, this axonal survival induces neuronal plasticity in reorganizing functional sensitive-motor loops networks. The reduction of hindpaw withdrawal thermal latency recordings in comparison to uninjured animals is also observed after penetrating SCI in both groups for all time points studied (Appendix A Appendix A).

Our two procedures also allow us to characterize the evolution of the lesion scar over time. Indeed, immunohistological results demonstrate that, in both cases, at 15 days after SCI, the scar is composed of a diffuse (after contusive SCI) or a dense (after penetrating SCI) fibrotic core. At this time point, there are only rare cystic cavities. In contrast, 60 days after SCI, in both cases, there is no or very few fibrotic scars; however, the spinal cord presents very large cystic cavities. These results are especially interesting because the lesion scar in rats 15 days after SCI is comparable to the one presents in mice [22,23]. At 60 days after SCI, the two scars are very different due to the fact that the fibrosis present in rats degenerates to the detriment of the cystic cavities, and this is not the case in mouse models. It appears that further studies will be necessary to understand the mechanisms responsible for this distinct lesion scar evolution over time. In a broader perspective, it was recently shown that this fibrotic scar is present in different central-nervous-system lesions in mice, such as traumatic brain or spinal cord injuries, as well as after demyelinating or ischemic lesion models [24]. It could also be interesting to demonstrate if this fibrotic scar is present throughout species after CNS lesions.

Recent studies have underlined the major role played by microglia/macrophages after SCI [25,26]. That is why we have investigated the effects of rTSMS treatment on these cellular populations. To do so, Iba1-staining experiments were performed. In particular, Iba1 intensity was measured, and, in both procedures and both time points investigated, it appears that there is no difference between groups. To complete this analysis, we quantified the amyboid-Iba1-positive cells’ areas present in the lesion epicenter. These analyses show that, in penetrating SCI (Procedure 1), rTSMS treatment decreases the amount of amyboid-Iba1-positive cell areas at both time points. In contrast, in contusive SCI (Procedure 2), the analyses show that rTSMS treatment increases the amount of the amyboid-Iba1-positive cells’ areas at 60 days after SCI. The amyboid shape quantification is usually employed to characterize the activation of microglia and macrophages [27]. Altogether, our results underline the fact that rTSMS treatment modulates microglia/macrophages activation and inflammatory response. However, the opposite results on Iba1-positive cells in our two procedures do not allow us to conclude about the specific effects of rTSMS on these cellular populations. Recent studies have demonstrated that circulating macrophages are mostly present in the lesion core, whereas microglia are present at the border of it [25]. It has also been shown that macrophages, instead of microglia, are mainly implicated in secondary axonal dieback [28]. Nevertheless, we can hypothesize that the inflammatory processes are different in our two procedures. Indeed, in Procedure 1, meninges are removed during the surgical procedure, whereas, in Procedure 2, meninges are still present after the surgery. We can assume that the ratio between microglia and macrophages varies in these two models. However, Iba1 staining is not specific and does not allow us to differentiate these two cellular populations into the lesion scar. In effect, we can hypothesize that, after penetrating SCI, the rupture of the meninges leads to a massive infiltration of circulating macrophages, and this should not be the case after contusive SCI, where the meninges are not removed. Thus, our results can reflect the positive effects of rTSMS on macrophages after penetrating SCI. Indeed, the immunomodulatory role played by magnetic stimulation (MS) on macrophages in culture has been already investigated [29]. It appears that MS can act on macrophages polarization. In contrast, after contusive SCI, the ratio between macrophages and microglia into the lesion scar should be different due to the reduction of macrophages infiltration. However, the precise role of MS and especially rTSMS on microglia is not clearly described. Moreover, as mentioned above, microglia and macrophages cannot be differentiated easily by immunostaining. Furthermore, the polarization of these inflammatory cells (M1 and M2) is also hard to conduct due to the possible shared expression of markers. That is why it could be of primary interest to conduct further investigations regarding the polarization of microglia/macrophages after rTSMS treatment. First, in vitro investigations based on well-characterized microglia and macrophages cultures could be performed. Then, single-cell sequencing could be a useful technic to reveal the diversity of the inflammatory cells present in the lesioned spinal cord, with or without rTSMS treatment. Some recent studies have pointed out the fact that there are specific microglial subpopulations associated with neurodegenerative diseases [30,31]. This kind of studies could be performed in order to characterize the role played by rTSMS on microglia/macrophages and, by extension, on the other inflammatory cells.

The main aim of the present research was to propose a preclinical study based on two complementary SCI models. Thus, its main limitation is that our study did not dissect the cellular and molecular mechanisms, and this can explain the role played by rTSMS after SCI. However, our study proposes that rTSMS exerts positive effects after SCI in rats in an animal model in which the consequences of SCIs are closer to the ones observed in humans. This study is a further step towards the use of this treatment in humans after SCI.

## Figures and Tables

**Figure 1 biomedicines-09-01827-f001:**
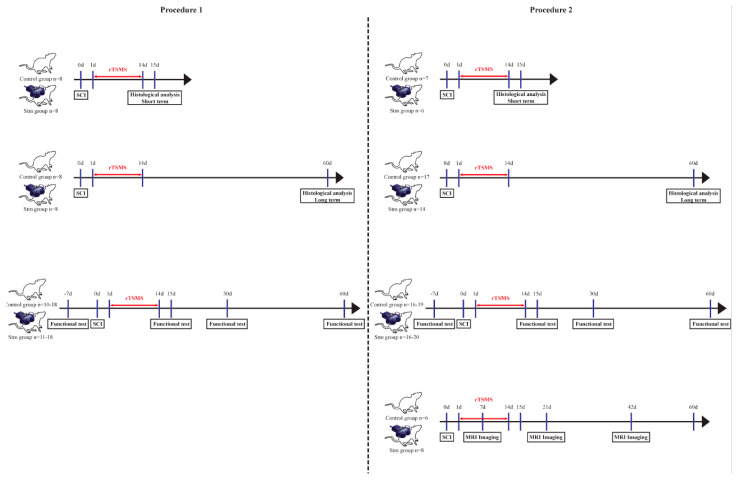
Experimental paradigms illustrating the timelines of the major experimental manipulations.

**Figure 2 biomedicines-09-01827-f002:**
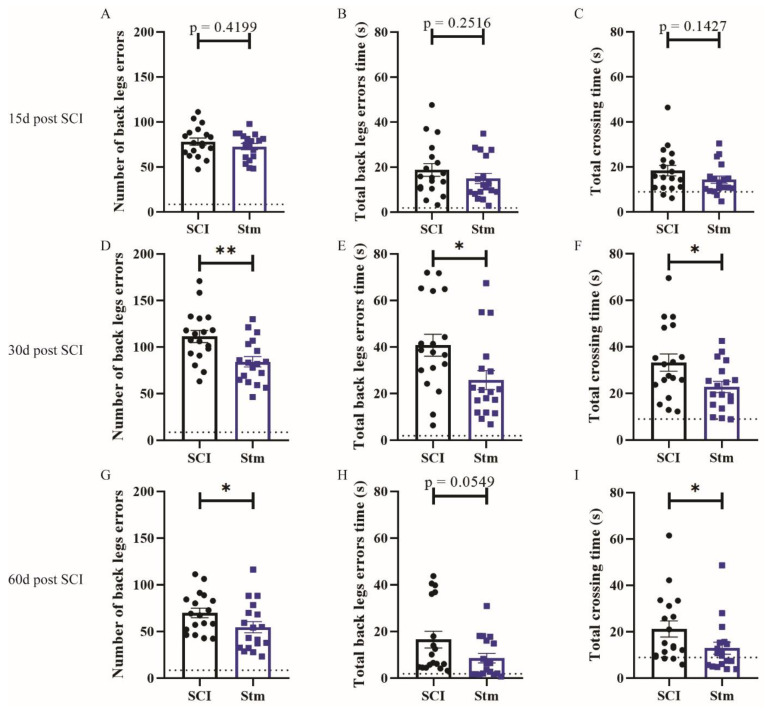
rTSMS treatment induces locomotor recovery after a complete transection of the spinal cord in rats. Quantification of locotronic evaluation at (**A**–**C**) 15 days, (**D**–**F**) 30 days and (**G**–**I**) 60 days after SCI. Parameters are (**A**,**D**,**G**) number of back-legs errors, (**B**,**E**,**H**) total back-legs errors’ time and (**C**,**F**,**I**) total crossing time. Quantifications are expressed as average ± SD; *n* = 18 animals per group. Dashed lines correspond to the baseline parameters obtained during locotronic habituation (7 days before SCI). Quantifications are expressed as average ± SD. Statistical evaluations are based on the Mann–Whitney test (* = *p* < 0.05 and ** = *p* < 0.01).

**Figure 3 biomedicines-09-01827-f003:**
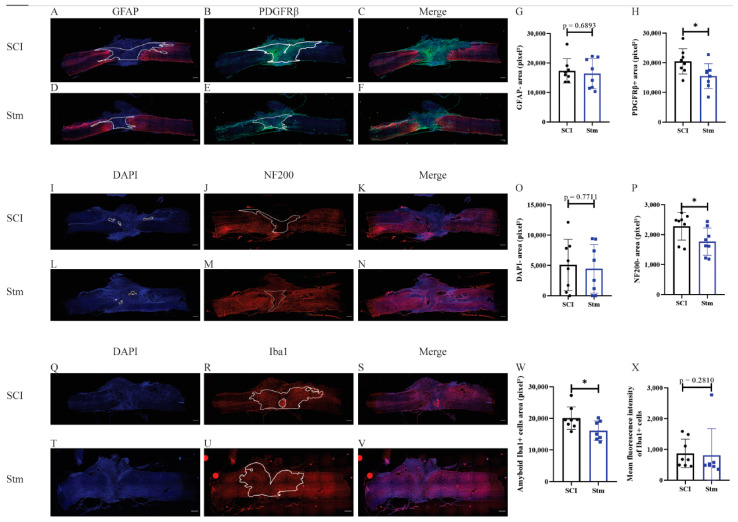
rTSMS treatment enhances tissue repair after a complete transection of the spinal cord 15 days after SCI in rats. At day 15, immunohistological analyses were performed. (**A**–**F**,**I**–**N**,**Q**–**V**) Representative pictures of sagittal spinal cord sections of (**A**–**C**,**I**–**K**,**Q**–**S**) SCI and (**D**–**F**,**L**–**N**,**T**–**V**) Stm (rTSMS treated) animals. Sections were stained with (**A**,**D**) GFAP, (**B**,**E**) PDGFRβ, (**I**,**L**,**Q**,**T**) DAPI, (**J**,**M**) NF200 and (**R**,**U**) Iba1. (**G**) Quantification of astrocytic-negative area (GFAP-). (**H**) Quantification of fibrosis-positive area (PDGFRβ+). (**O**) Quantification of DAPI-negative area (DAPI-). (**P**) Quantification of NF200-negative area (NF200-). (**W**) Quantification of Iba1-amyboid-positive cells’ area (Iba1+) and (**X**) quantification of Iba1+ mean fluorescence intensity. Scale bars are 200 µm; *n* = 8 animals per group. Quantifications are expressed as average ± SD. Statistical evaluations are based on the Mann–Whitney test (* = *p* < 0.05).

**Figure 4 biomedicines-09-01827-f004:**
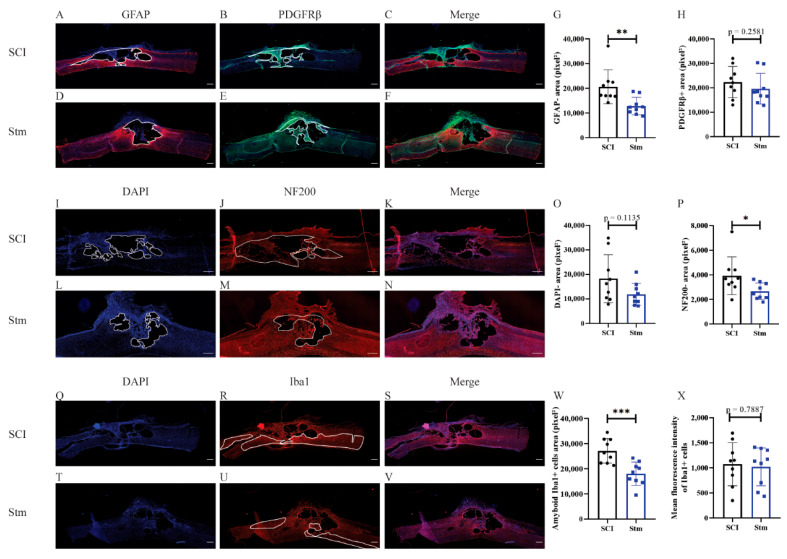
rTSMS treatment enhances tissue repair after a complete transection of the spinal cord 60 days after SCI in rats. At day 60, immunohistological analyses were performed. (**A**–**F**,**I**–**N**,**Q**–**T**,**U**,**V**) Representative pictures of sagittal spinal cord sections of (**A**–**C**,**I**–**K**,**Q**–**S**) SCI and (**D**–**F**,**L**–**N**,**T**–**V**) Stm (rTSMS treated) animals. Sections were stained with (A and D) GFAP, (B and E) PDGFRβ, (**I**,**L**,**Q**,**T**) DAPI, (**J**,**M**) NF200 and (**R**,**U**) Iba1. (**G**) Quantification of astrocytic-negative area (GFAP-). (**H**) Quantification of fibrosis-positive area (PDGFRβ+). (**O**) Quantification of DAPI-negative area (DAPI-). (**P**) Quantification of NF200-negative area (NF200-). (**W**) Quantification of Iba1 amyboid-positive cells’ area (Iba1+) and (**X**) quantification of Iba1+ mean fluorescence intensity. Scale bars are 200 µm; *n* = 8 animals per group. Quantifications are expressed as average ± SD. Statistical evaluations are based on the Mann–Whitney test (* = *p* < 0.05, ** = *p* < 0.01 and *** = *p* < 0.001).

**Figure 5 biomedicines-09-01827-f005:**
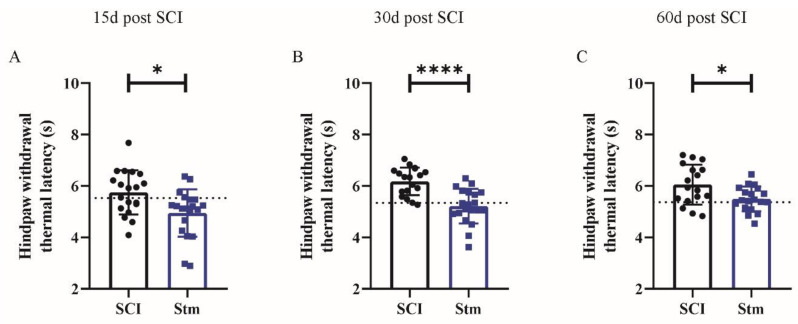
rTSMS treatment modulate sensitive recovery after SCI in rats. Quantification of hindpaw withdrawal thermal latency at (**A**) 15 days, (**B**) 30 days and (**C**) 60 days after SCI. Quantifications are expressed as average ± SD; *n* = 19 animals per SCI group and *n* = 20 animals per STM group at 15 days, *n* = 18 animals per SCI group and *n* = 20 animals per STM group at 30 days and *n* = 17 animals per SCI group and *n* = 19 animals per Stm group 60 days after SCI. Dashed lines correspond to the baseline parameters obtained with non-injured animals. Statistical evaluations were based on the Mann–Whitney test (* = *p* < 0.05 and **** = *p* < 0.0001).

**Figure 6 biomedicines-09-01827-f006:**
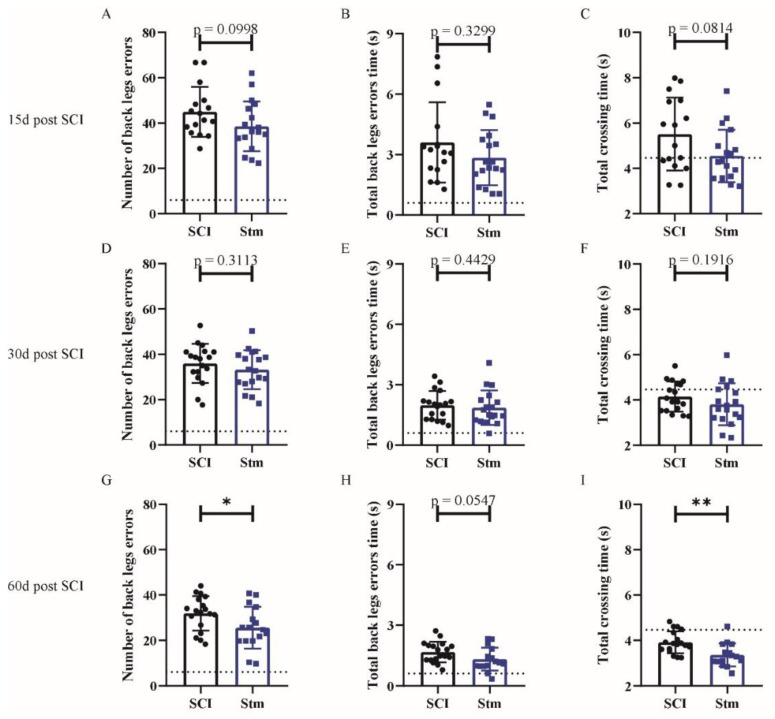
rTSMS treatment induces locomotor recovery after contusive SCI in rats. Quantification of locotronic evaluation at (**A**–**C**) 15 days, (**D**–**F**) 30 days and (**G**–**I**) 60 days after SCI. Parameters are (**A**,**D**,**G**) number of back-leg errors, (**B**,**E**,**H**) total back-leg errors’ time and (**C**,**F**,**I**) total crossing time. Quantifications are expressed as average ± SD; *n* = 16 animals per SCI group and *n* = 18 animals per Stm group at 15 days, *n* = 18 animals per SCI group and *n* = 18 animals per Stm group at 30 days and *n* = 18 animals per SCI group and *n* = 16 animals per Stm group 60 days after SCI. Dashed lines correspond to the baseline parameters obtained during locotronic habituation (7 days before SCI). Quantifications are expressed as average ± SD. Statistical evaluations are based on the Mann–Whitney test (* = *p* < 0.05 and ** = *p* < 0.01).

**Figure 7 biomedicines-09-01827-f007:**
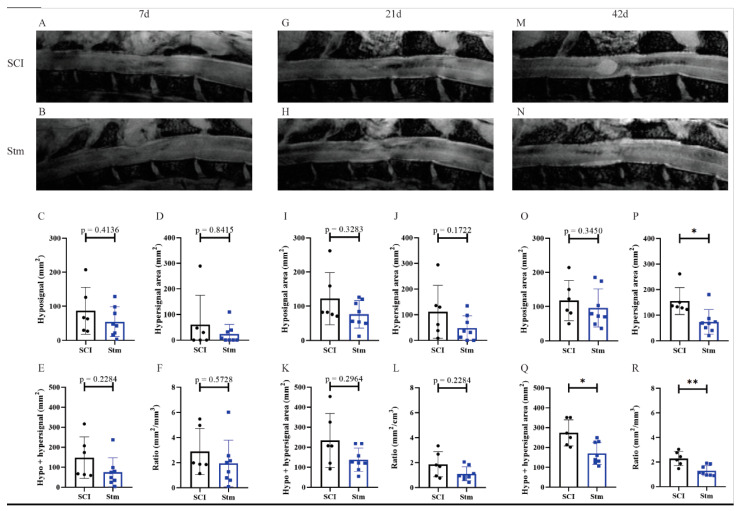
MRI analyses show that rTSMS treatment decreases cystic cavities and increases spinal cord spared tissue. At (**A**–**F**) 7, (**G**–**L**) 21 and (**M**–**R**) 42 days, MRI experiments were performed. Representative images of sagittal spinal cord sections recorded with MRI of (**A**,**G**,**M**) SCI and (**B**,**H**,**N**) Stm (rTSMS treated) animals. Quantification of hyposignal area (**C**) 7, (**I**) 14 and (**O**) 42 days after SCI. Quantification of hypersignal area (**D**) 7, (**J**) 14 and (**P**) 42 days after SCI. Quantification of hypo + hypersignal area (**E**) 7, (**K**) 14 and (**Q**) 42 days after SCI. Quantification of ratio lesioned tissue (**F**) 7, (**L**) 14 and (**R**) 42 days after SCI; *n* = 6 animals per control group and *n*= 8 per Stm group. Quantifications are expressed as average ± SD. Statistical evaluations are based on the Mann–Whitney test (* = *p* < 0.05 and ** = *p* < 0.01).

**Figure 8 biomedicines-09-01827-f008:**
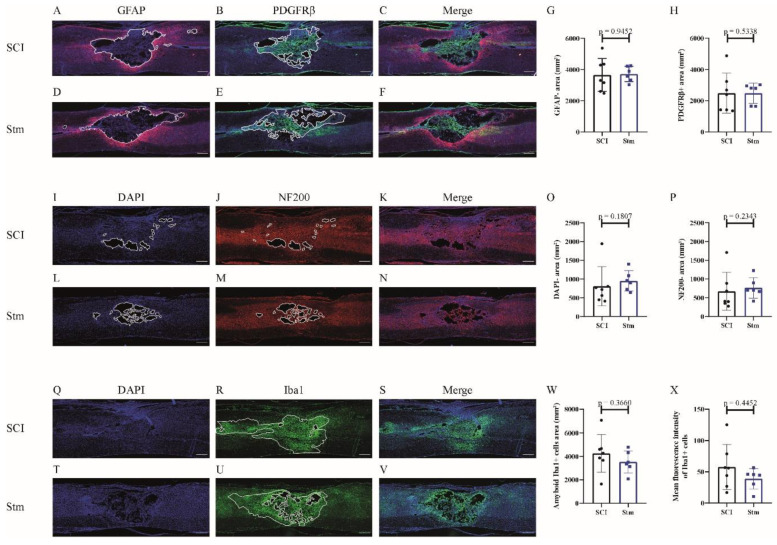
rTSMS treatment does not have effect on tissue repair after contusive SCI in rats 15 days after SCI. At day 15, immunohistological analyses were performed. (**A**–**V**) Representative pictures of sagittal spinal cord sections of (**A**–**C**,**I**–**K**,**Q**–**S**) SCI and (**D**–**F**,**L**–**N**,**T**–**V**) Stm (rTSMS treated) animals. Sections were stained with (**A**,**D**) GFAP, (**B**,**E**) PDGFRβ, (**I**,**L**,**Q**,**T**) DAPI, (**J**,**M**) NF200 and (R and U) Iba1. (**G**) Quantification of astrocytic-negative area (GFAP-). (**H**) Quantification of fibrosis-positive area (PDGFRβ+). (**O**) Quantification of DAPI-negative area (DAPI-). (**P**) Quantification of NF200-negative area (NF200-). (**W**) Quantification of Iba1-amyboid-positive cells’ area (Iba1+) and (**X**) quantification of Iba1+ mean fluorescence intensity. Scale bars are 200 µm; *n* = 7 animals per SCI group and *n* = 6 animals per Stm group. Quantifications are expressed as average ± SD. Statistical evaluations are based on the Mann–Whitney test.

**Figure 9 biomedicines-09-01827-f009:**
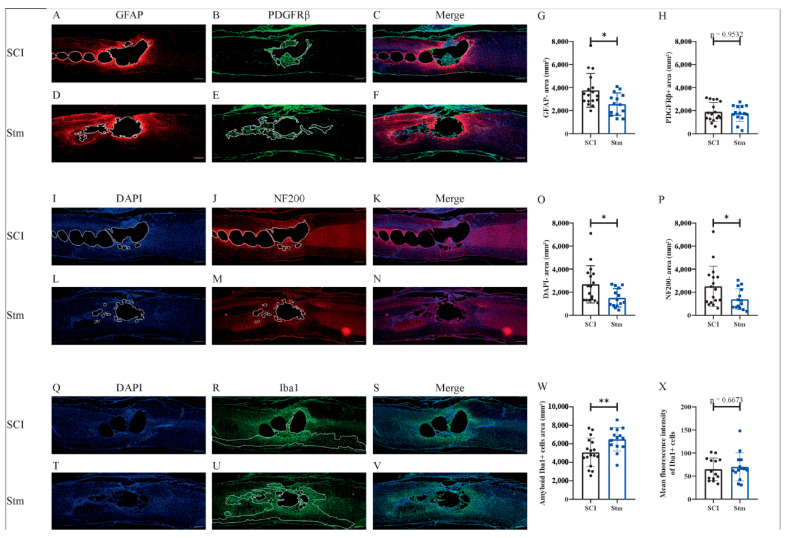
rTSMS treatment enhances tissue repair after contusive SCI in rats 60 days after SCI. At day 60, immunohistological analyses were performed. (**A**–**F**,**I**–**N**,**Q**–**V**) Representative pictures of sagittal spinal cord sections of (**A**–**C**,**I**–**K**,**Q**–**S**) SCI and (**D**–**F**,**L**–**N**,**T**–**V**) Stm (rTSMS treated) animals. Sections were stained with (**A**,**D**) GFAP, (**B**,**E**) PDGFRβ, (**I**,**L**,**Q**,**T**) DAPI, (**J**,**M**) NF200 and (**R**,**U**) Iba1. (**G**) Quantification of astrocytic-negative area (GFAP-). (**H**) Quantification of fibrosis-positive area (PDGFRβ+). (**O**) Quantification of DAPI-negative area (DAPI-). (**P**) Quantification of NF200-negative area (NF200-). (**W**) Quantification of Iba1-amyboid-positive cells’ area (Iba1+) and (**X**) quantification of Iba1+ mean fluorescence intensity. Scale bars are 200 µm; *n* = 17 animals per SCI group and *n* = 14 per Stm group. Quantifications are expressed as average ± SD. Statistical evaluations are based on the Mann–Whitney test (* = *p* < 0.05 and ** = *p* < 0.01).

**Table 1 biomedicines-09-01827-t001:** Summary table of the acquisition parameters of MRI sequences used to characterize and identify spinal cord structure on rats after SCI and/or rTSMS treatment.

	TR (ms)	TE (ms)	Thickness (µm)	FOV (cm)	Matrix (pixel)	Acquisition Time
T2 *-weighted axial section	2000	6.5 20.4 34.3 48.1 62.1 75.9 89.1 103.7 117.5 131.4	500	4	256	15 min 56 s
T2 *-weighted sagittal section	2000	6.5	500	4	320	16 min

## Data Availability

The authors confirm that the data supporting the findings of this study are available within the article and its Appendix A.

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
