# Peer review of "Repetitive Trans Spinal Magnetic Stimulation Improves Functional Recovery and Tissue Repair in Contusive and Penetrating Spinal Cord Injury Models in Rats"

_biomedicines, 2021, doi:10.3390/biomedicines9121827_

Round 1
Reviewer 1 Report
In this report the authors proposed repetitive trans spinal magnetic
stimulation (rTSMS) as a therapeutic approach for spinal cord tissue
repair after injury in rats. The authors examined the spinal cord tissue
recovery by functional (plantar and locotronic), MRI and immunohistological
tests. The authors found that rTSMS induces functional recovery of
spinal cord after injury. The authors have previously reported similar
findings in mice (10.1007/s13311-020-00915-5).
Author Response
Response: First of all, we would like to thank the reviewer for his comments.
Reviewer 2 Report
In this manuscript, Robac et al characterize the repetitive trans-spinal magnetic stimulation treatment to improve functional recovery and tissue repair in contusive and penetrating spinal cord injury rat model.
The authors claim that rTSMS treatment enhances tissue repair, decreases cystic cavities and increases spinal cord spared tissue after contusive SCI in rats 60 days after SCI. The design is well done and sufficient amount of animals is provided giving the expectable difficulty to encounter a lot of variation between animals. However there are some points that are not clear for the reviewer:
In figure 4 the authors sometimes shows the dotted regions of lesion marking the lesions focusing in the holes of death regions (such as panels A, D, I, L). However, in the same figure, other panels mark regions with plenty of staining without taking into account the same criteria (panels B, E, R, U). Why? if the cell counts are referenced to these regions, must be identical (or at least following always the same criteria).
The authors only compare all the time points 15, 30 and 60 days in graphs of figures 2, 5, 6 and the supplemental figure 1. However no images of 30 days are provided, focusing the majority of the data in the time-point of 15 days (Fig. 3&8) and 60 days (Fig. 4&9), due to all quantifications are done, the reader expect to see at least how looks like the lesion and the treatment at 30 day post-lesion.
The authors should discuss more in depth how can explain to have an increased tissue repair at 60 days with an increased number of inflammatory Iba1+ cell population.
Minor points:
Please, check the underlines of the figure legend "Figure 9._rTSMS", "Figure 6._" and the underline of "Figure 7. ". Please use always the same criteria and style (the other figures does not have underline).
Author Response
In this manuscript, Robac et al characterize the repetitive trans-spinal magnetic stimulation treatment to improve functional recovery and tissue repair in contusive and penetrating spinal cord injury rat model.
The authors claim that rTSMS treatment enhances tissue repair, decreases cystic cavities and increases spinal cord spared tissue after contusive SCI in rats 60 days after SCI. The design is well done and sufficient amount of animals is provided giving the expectable difficulty to encounter a lot of variation between animals. However there are some points that are not clear for the reviewer:
Response: First of all, we would like to thank the reviewer for his comments and questions.
1 - In figure 4 the authors sometimes shows the dotted regions of lesion marking the lesions focusing in the holes of death regions (such as panels A, D, I, L). However, in the same figure, other panels mark regions with plenty of staining without taking into account the same criteria (panels B, E, R, U). Why? if the cell counts are referenced to these regions, must be identical (or at least following always the same criteria).
Response: the dotted regions in A, D, I and L are related to areas without staining. Indeed, in A and D, the dotted regions represent the GFAP negative areas, the quantification of the lack of GFAP is in G. For I and L, the dotted regions represent the areas without DAPI. These regions are the cavities (without cells) and the quantification for these areas are in O.
At the opposite, the dotted regions in B, E, R and U represent areas with staining. In B and E, the dotted regions represent the fibrotic areas which express PDGFβ. The quantification for these areas is in H. For R and U, the dotted regions represent the amyboid positive cells areas, the regions in which we can find activated microglia/macrophages cells. The quantification for these regions is in W.
2 - The authors only compare all the time points 15, 30 and 60 days in graphs of figures 2, 5, 6 and the supplemental figure 1. However no images of 30 days are provided, focusing the majority of the data in the time-point of 15 days (Fig. 3&8) and 60 days (Fig. 4&9), due to all quantifications are done, the reader expect to see at least how looks like the lesion and the treatment at 30 day post-lesion.
Response: We agree with reviewer’s remark. It could have been very informative to add an intermediate time point, 30 days after SCI. However, we already performed SCI on 112 rats and 104 rats have been included in our study. That is why, we did not add this specific time point. Our results indicate that rTSMS based treatment induces an effect on tissue repair 15 days after penetrating SCI whereas rTSMS has a positive effect on tissue repair only 60 days after contusive SCI. That is why, we have added MRI analyses after contusive SCI. These experiments have been performed 7, 21 and 42 days after SCI, they reveal that rTSMS significantly modulates the lesion scar 42 days after SCI.
3 - The authors should discuss more in depth how can explain to have an increased tissue repair at 60 days with an increased number of inflammatory Iba1+ cell population.
Response: We agree with reviewer’s comment. These specific results are very interesting and have been discussed more in depth in the revised version of the manuscript.
Minor points:
Please, check the underlines of the figure legend "Figure 9._rTSMS", "Figure 6._" and the underline of "Figure 7. ". Please use always the same criteria and style (the other figures does not have underline).
Response: The manuscript has been checked for spelling, grammar and typo.
Round 2
Reviewer 2 Report
The authors fully addressed all my concerns.